# Immunogenic Cell Death (ICD)-Related Gene Signature Could Predict the Prognosis of Patients with Diffuse Large B-Cell Lymphoma

**DOI:** 10.3390/jpm12111840

**Published:** 2022-11-04

**Authors:** Liqin Ping, Yanxia He, Yan Gao, Xiaoxiao Wang, Cheng Huang, Bing Bai, Huiqiang Huang

**Affiliations:** State Key Laboratory of Oncology in South China, Department of Medical Oncology, Sun Yat-sen University Cancer Center, Collaborative Innovation Center for Cancer Medicine, Guangzhou 510060, China

**Keywords:** immunogenic cell death, diffuse large B-cell lymphoma, ICD, DLBCL, GEO, prognosis

## Abstract

Background: Diffuse large B-cell lymphoma (DLBCL) is the most prevalent type of lymphoma that is potentially curable by chemotherapy. Immunogenic cell death (ICD) is regarded as an essential process for the clearance of residual tumor cells. However, the impact of ICD on DLBCL remains unknown. Here, we tried to explore the prognostic role of ICD in DLBCL. Methods: A gene expression microarray of DLBCL was downloaded from the Gene Expression Omnibus (GEO). The genes involved in ICD were obtained via literature reviews. Then, based on univariate, multivariate, and LASSO Cox regression analysis, the ICD-related gene signature was identified. The effect of the ICD-related gene signature on DLBCL was explored. The chi-square test was used to compare complete response rate (CRR) and recurrence rate between high- and low-risk groups. Results: The signature based on 12 ICD-related genes could independently predict the overall survival of DLBCL. Furthermore, high risk was linked to lower CRR and higher recurrence rate. Then, a nomogram based on the ICD-related gene signature was established. The area under the curve of the prediction model reached 0.820 in the training set and 0.780 in the validation set. Conclusions: This study suggested that the ICD-related gene signature could be a novel prognostic indicator for DLCBL.

## 1. Background

Diffuse large B-cell lymphoma (DLBCL) is the most prevalent lymphoma subtype, having apparent heterogeneity [1,2]. The 5-year overall survival (OS) rate could reach 60%; however, up to 50% of patients experienced disease progression [3]. The accurate prediction of the prognosis, response to chemotherapy, and recurrence after achieving complete response (CR) is vital for guiding the treatment regimen and follow-up frequency for DLBCL patients. The commonly used International Prognostic Index (IPI) is based on the clinical characteristics, while molecular characteristics are ignored, which has limited predictive power [4]. Additionally, the IPI cannot guide the selection of treatment regimen. Thus, it is critical to identify the molecular characteristics of DLBCL patients, which can predict the prognosis, response to chemotherapy, and risk of relapse.

Immunogenic cell death (ICD) is a stress-driven cell death regulated by a series of genes [5]. ICD can enhance the antigenicity and adjuvanticity of dying tumor cells, thus activating the anti-tumor immunity [6,7]. In addition to pathogens [8], chemotherapeutic drugs are important inducers that can trigger ICD [9]. The common drugs for treating DLBCL, such as cyclophosphamide [10] and doxorubicin [11], are essential factors that can induce ICD. After chemotherapy, the cell membrane surfaces of dying tumor cells expose calreticulin (CALR) and heat shock proteins (HSPs) and then release DAMPs [12]. The latter bind receptors on the surfaces of dendritic cells (DCs), thus promoting the phagocytosis of DCs and the presentation of tumor-associated antigens (TAAs) [7]. Therefore, treated with chemotherapeutic agents, tumor cells undergo gene-driven programmed cell death. Then, a robust antigen-specific immune reaction is triggered to clear the remaining tumor cells, thus reducing the risk of tumor recurrence [13,14].

DLBCL is one of the malignant tumors that are potentially curable by chemotherapy, implying that ICD may be critical for DLBCL. Some patients who achieved CR after chemotherapy could obtain long-term survival, although others experienced relapse. Therefore, we hypothesize that ICD-related genes are deeply associated with the prognosis and response to chemotherapy in DLBCL patients. At present, there is much research focus on ICD-related genes, but no systematic ICD-related gene signature has been established. This study aims to investigate the role of ICD-related genes in DLBCL.

## 2. Materials and Methods

### 2.1. Collection of Data

The transcriptional information and clinical information on the DLBCL patients were downloaded from Gene Expression Omnibus (GEO, http://www.ncbi.nlm.nih.gov/geo/) on 1 May 2021. Both GSE31312 and GSE10846 were publicly available in the GEO database, and the limma package of “R” was used to standardize the transcriptional data of the two databases. Patients with missing clinical information including age, number of extranodal lesions, Eastern Cooperative Oncology Group (ECOG) score, Ann Arbor stage, cell of origin (COO) classification, lactate dehydrogenase (LDH) status, and survival time and status were excluded. The first-line chemotherapy regimen was similar in the two datasets; patients were treated with a CHOP-like regimen with or without rituximab. According to the relevant reviews reported on ICD [5,7,9,12], we collected the genes involved in the process of ICD and analyzed these genes in this study.

### 2.2. Identification and Validation of ICD-Related Gene Signature

Patients from the GSE31312 database were used as the training group to construct the gene signature, while the GSE10846 database was used as the validation group. The candidate genes for the ICD-related gene signature were identified based on the univariate and multivariate Cox regression analysis of GSE31312. Subsequently, least absolute shrinkage and selection operator (LASSO) Cox regression analysis was performed on all candidate genes to obtain the optimal gene signature. Furthermore, for each patient, the risk score was determined through gene expression and the corresponding regression coefficient. The risk score for ICD-related gene signature was calculated utilizing the formula risk score = e ^ sum (level of expression × corresponding regression coefficient). Patients in the training group were divided into high-risk and low-risk groups on the basis of the median risk score. Using the same cut-off, patients in the validation set were separated into low- and high-risk groups. Kaplan–Meier analysis was applied to compare the survival between the high-risk and low-risk groups. The independent prognostic importance of the risk score was investigated with Cox regression analysis.

### 2.3. Relationship between Risk Score and Clinical Parameters

The chi-square test was performed to analysis the relationships between the risk score and clinical factors (such as age, gender, ECOG score, Ann Arbor stage, extranodal lesions, and COO classification). Considering that the GSE10846 validation set lacked information on treatment response, the association between risk score and chemotherapy efficacy was only analyzed in the training set, GSE31312.

### 2.4. GSEA (Gene Set Enrichment Analysis)

The GSE31312 RNA-seq data were subjected to gene set enrichment analysis (GSEA; http://www.broadinstitute.org/gsea) on 17 May 2021 for investigating the molecular and biological dissimilarities among the low- and high-risk patients. The gene sets were filtered using the minimum and maximum gene set sizes of 15 and 500 genes, respectively.

### 2.5. Statistical Analysis

SPSS (v.24.0, IBM Corp., Armonk, New York, USA) and R. (v.4.0.3, R Statistical Computing Foundation, Vienna, Austria) software were utilized for the statistical analysis. To investigate the connections between risk group, clinical features, and treatment response, the chi-square test was applied. Univariate and multivariate Cox regression analysis was applied to investigate overall survival (OS) prognostic variables. The optimal gene signature was identified using LASSO Cox regression analysis. The nomogram model including clinical features and risk score was constructed using the ‘rms’ package in R software. Receiver operating characteristic (ROC) curve analysis was used to evaluate the nomogram’s prediction accuracy. The statistical significance was defined as *p* less than 0.05, and *p* value was two-tailed.

## 3. Results

### 3.1. DLBCL Patients’ Clinical Features

Due to inadequate information, 83 patients in GSE31312 and 100 patients in GSE10846 were excluded. A total of 415 patients (GSE31312) were utilized as the training set in this study, while 320 patients (GSE10846) were used as the validation set. The study design and flowchart of this study are displayed in Figure 1. The clinical features of GSE31312 and GSE10846 are summarized in Table 1.

### 3.2. Construction and Verification of ICD-Related Gene Signature

Through literature reviews, a total of 108 ICD-related genes were identified (Appendix A). The univariate Cox regression analysis of the training set revealed that 13 ICD-related genes were related to OS in DLBCL patients (Figure 2A). Then, the LASSO Cox regression analysis was performed to identify the 12 optimal genes for the ICD-related gene signature (Figure 2B,C). The risk score of ICD-related gene signature = e ^ sum (0.441468766197694 × CALR-0.0386737231994825 × IFNG − 0.0141607246226577 × EIF2A − 0.0516818356615812 × IL6ST + 0.887428533133031 × IL17A − 0.197026941268847 × TNF − 0.0739231129957232 × PDIA3 + 0.496266758690389 × BAK1 -0.0786254627760117 × TLR4 − 0.291982828290371 × CTLA4 − 0.263952067985493 × HLA.DQB2 − 0.609604119510768 × ERAP1). The training set was separated into high-risk (*n* = 207) and low-risk (*n* = 208) groups on the basis of the median risk score (Figure 3A). Patients in the high-risk group had higher risk of death than those in the low-risk group (Figure 3C). According to the Kaplan–Meier analysis, the OS of DLBCL patients in the high-risk group was shorter than the low-risk group (Figure 3E, *p* < 0.001). The validation set’s patients were then classified as low or high risk using the same cut-off as the training set (Figure 3B,D). Patients with high risk in the validation group also had poorer prognosis (Figure 3F, *p* < 0.001).

### 3.3. Univariate and Multivariate Cox Regression Analysis of Potential Prognostic Factors

Potential prognostic factors, including ICD-based risk score, gender, age, Ann Arbor stage, ECOG score, COO classification, number of extranodal lesions, and LDH status, were included in the univariate and multivariate Cox regression analysis. The univariate analysis showed that ICD-based risk score could predict the OS of DLBCL in both the training set (HR = 2.646, 95% CI: 1.859–3.766, *p* < 0.001) and the validation set (HR = 2.360, 95% CI: 1.629–3.419, *p* < 0.001) (Figure 4A,B). The multivariate Cox regression analysis showed that ICD-based risk score was an independent prognostic factor for DLBCL in both the training set (HR = 2.503, 95% CI: 1.736–3.610, *p* < 0.001) and the validation set (HR = 2.368, 95% CI: 1.594–3.517, *p* < 0.001) (Figure 4C,D).

### 3.4. Higher CR Rate and Lower Recurrence Rate in Low-Risk Group

To assess the connections between ICD-based risk score and clinical factors, the chi-square test was performed. According to the analysis, those in the high-risk group had a higher percentage of non-GCB (germinal center B cells) subtype, and more than two extranodal lesions were involved (Table 2). The relationship between chemotherapy efficacy and ICD-based risk score was explored in the GSE31312 database, indicating that low risk was associated with higher CR rate and overall response rate (ORR) (Figure 5A). Furthermore, the high-risk group was linked to higher recurrence rate among those who achieved CR (Figure 5B).

### 3.5. Gene Set Enrichment Analysis (GSEA)

The biological functions and signaling pathways associated with the ICD-related gene signature were investigated using gene set enrichment analysis (GESA). The results showed that genes in the antitumor immunity pathways, such as antigen processing ubiquitination proteasome degradation, NOD 1/2 signaling, T cell receptor signaling, B cell receptor signaling, costimulation by the CD28 family, and TNF signaling, were upregulated in the low-risk DLBCL patients (Figure 6).

### 3.6. A Nomogram Based on ICD-Related Gene Signature

To improve the predictive effectiveness of the prediction model, the nomogram for DLBCL patients was established using biomarkers including ICD-based risk score and clinical features (Figure 7A). A high consistency in the calibration plots of the 3-year and 5-year survival was observed between observation and prediction (Figure 7B,C). According to the time-dependent ROC curve analysis, the areas under the curve (AUC) for the 2-year survival rate were 0.766; 3-year, 0.77; and 5-year, 0.784 (Figure 7D), whereas they were 0.75, 0.769, and 0.769 in the validation set, respectively (Figure 7E).

## 4. Discussion

DLBCL is a potentially curable malignancy with chemotherapy. After chemotherapy, a significant proportion of patients achieve CR. The prediction efficacy of the commonly used prognostic model of IPI is limited because it fails to include the molecular characteristics. ICD is a form of regulated cell death that is regulated by genes. The antigenicity of tumor cells is an important factor in the induction of ICD by chemotherapy [9]. DAMPs released during ICD act as powerful adjuvants for immune responses, promoting the recruitment and activation of immune cells [6,7]. ICD can promote anti-tumor immunity and is critical in clearing the residual tumor cells. Therefore, we hypothesized that ICD-related genes were connected with DLBCL treatment response and prognosis.

This study is the first to establish the ICD-related gene signature, which was an independent prognostic predictor for DLBCL patients. This signature provided a novel biomarker for a prognosis prediction model for DLBCL. At the same time, the ICD-related gene signature can predict the efficacy of chemotherapy and relapse after achieving CR, which could further guide the follow-up frequency. Compared with the IPI, the nomogram based on this signature improved the predictive performance of the prediction model.

In this study, CALR, IL17A, and BAK1 were associated with poor outcome. IFNG, EIF2A, IL6ST, TNF, PDIA3, TLR4, CTLA4, HLA.DQB2, and ERAP1 suggested longer OS for DLBCL patients. Dying tumor cells expose the CALR/ERp57 complex to the cell surface, triggering a powerful anticancer immune response [15]. The effect of CALR on the prognosis of malignant tumors was controversial. The expression of CALR was associated with the metastasis and invasion of bladder cancer, leukemia, and other malignant tumors [12]. The immunosuppressive tumor microenvironment induced by IL17 was associated with poorer prognosis [16]. Previous studies reported that BAK1 expression was related to poor prognosis for malignancies such as DLBCL [17,18], which is consistent with the present findings. EIF2A [17] and PDIA3 [15] participated in the CALR exposure process in ICD and promoted the immune response. HLA.DQB2 [18] and ERAP1 [19] are involved in producing HLA-II and HLA-I molecules, respectively, and take a crucial part in presenting extracellular antigens.

Through Cox regression analysis, the ICD-associated gene signature was identified as an independent prognostic predictor for DLBCL patients. Furthermore, the CRR and ORR were higher in patients with low risk. In the low-risk group, patients who achieved CR had lower recurrence rate, implying that ICD was critical in eliminating residual lesions and thereby lowering the recurrence rate.

However, this study is based on biological information analysis and has some limitations. The data used in this study were from the GEO public database. Although the results suggest that ICD-related gene signature can predict the prognosis and recurrence of DLBCL, it still needs to be verified in the real world in the future. Basic in vivo or in vitro experiments should be conducted to confirm the effect of ICD on DLBCL. 

## 5. Conclusions

In conclusion, our findings imply that the ICD-related gene signature might be used as an alternative approach for DLBCL, predicting OS and recurrence risk after CR. They call for further prospective studies to explore the role of the ICD-related gene signature in DLBCL.

## Figures and Tables

**Figure 1 jpm-12-01840-f001:**
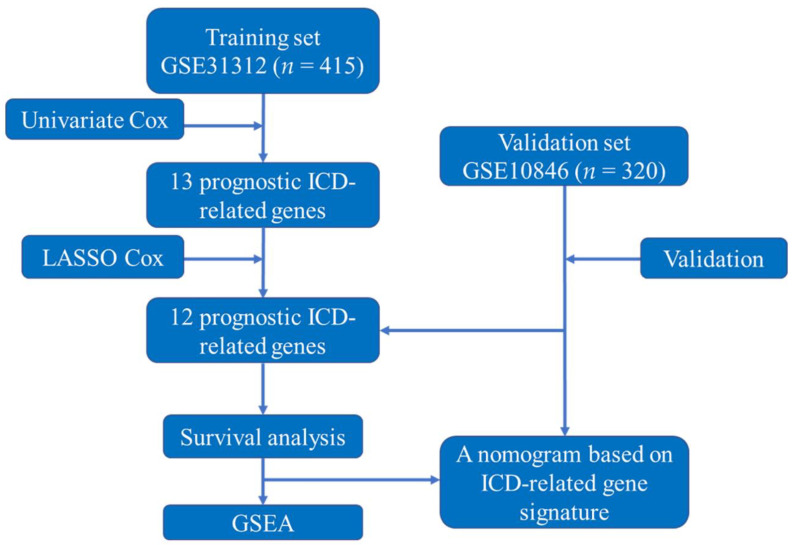
The study’s design and flowchart. Abbreviations: ICD, immunogenic cell death; LASSO, least absolute shrinkage and selection operator; GSEA, gene set enrichment analysis.

**Figure 2 jpm-12-01840-f002:**
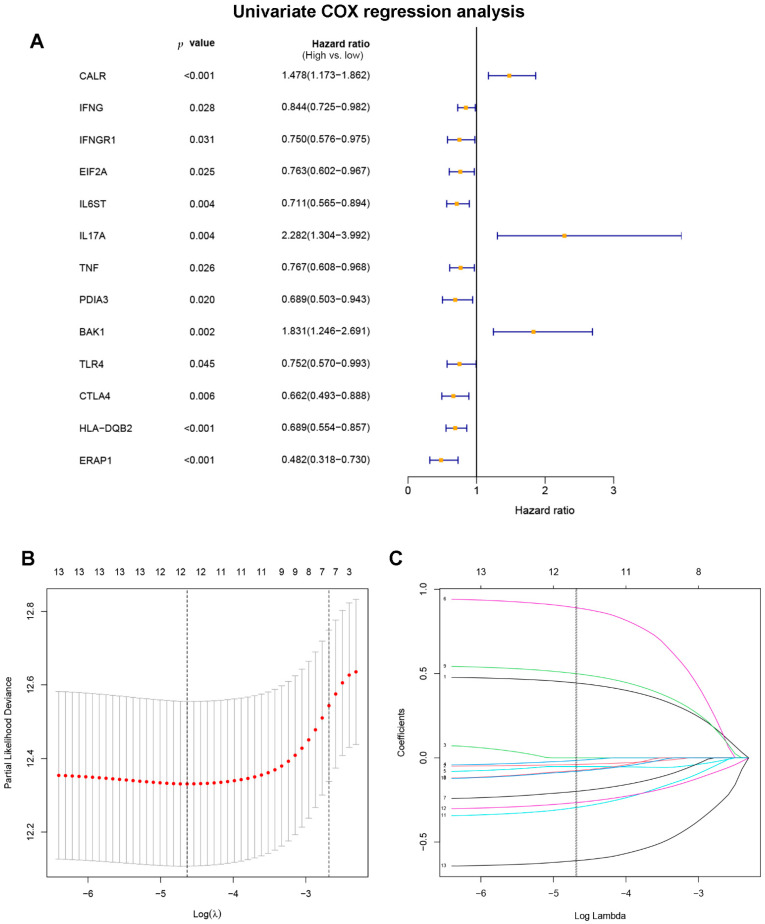
Construction of ICD-related gene signature. (**A**) Identify prognostic immunogenic cell death-related genes through univariate Cox regression analysis. (**B**) The optimal λ was selected, which was drawn with imaginary perpendicular lines. (**C**) LASSO Cox regression analysis revealed twelve immunogenic cell death-related genes.

**Figure 3 jpm-12-01840-f003:**
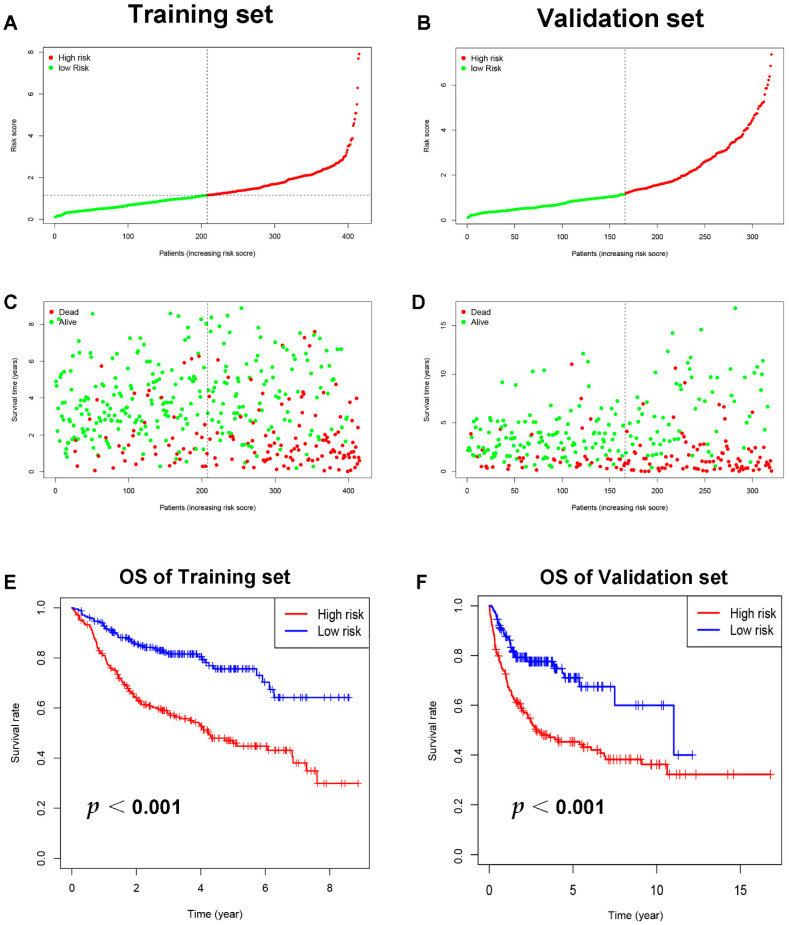
The training and validation sets’ relationships between ICD-related gene signature and prognosis. (**A**) The training set distribution of risk score and (**B**) that for the validation set. Survival status, overall survival, and distributions of training set risk score (**C**) and validation set risk score (**D**). Kaplan–Meier graphs illustrate the differences in OS among low-risk and high-risk groups in both the training (**E**) and validation sets (**F**). Abbreviations: OS, overall survival.

**Figure 4 jpm-12-01840-f004:**
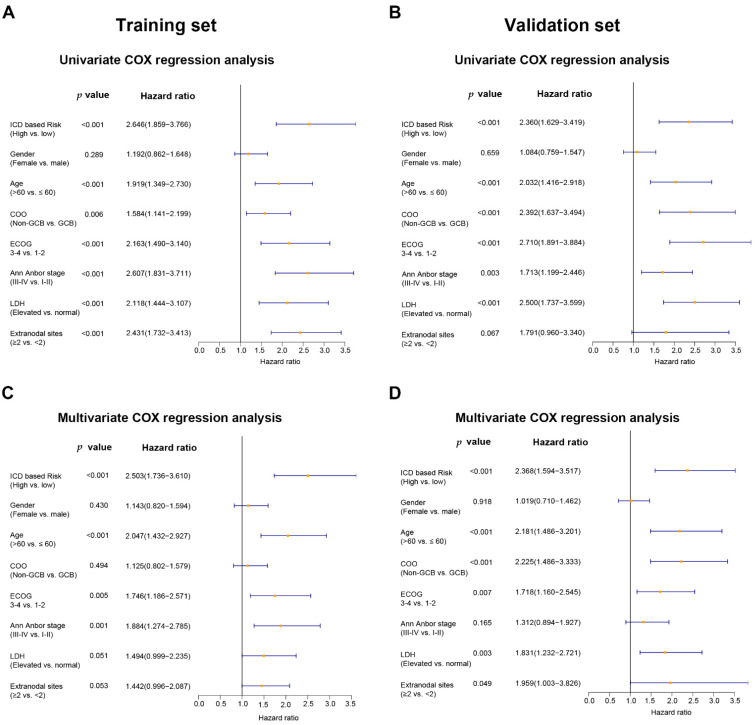
For DLBCL patients, the ICD-related gene signature is an independent predictive factor. The multivariate and univariate Cox regression analyses of OS calculated in the training set (**A**,**C**) and the validation set (**B**,**D**). Abbreviations: COO, cell of origin; GCB, germinal center B cells; ECOG, Eastern Cooperative Oncology Group; LDH, lactate dehydrogenase.

**Figure 5 jpm-12-01840-f005:**
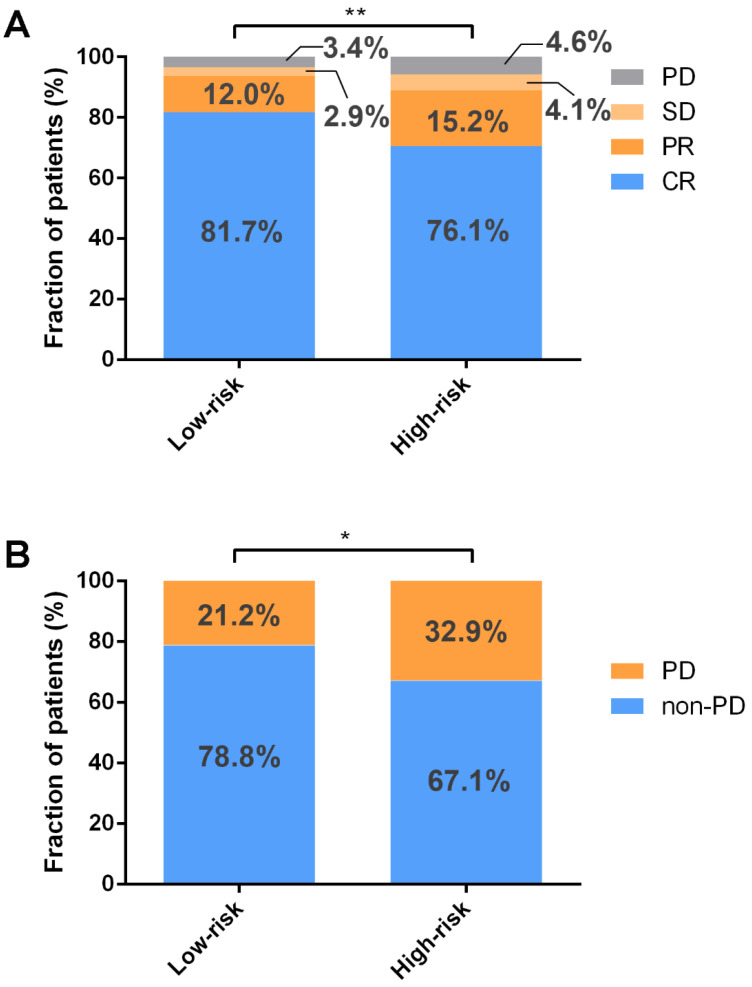
ICD-related gene signature is associated with chemotherapy response. (**A**) Low risk is associated with higher rates of CR and ORR by Wilcoxon rank–sum test. (**B**) The chi-square test found that among patients who achieved CR, the high-risk group had a greater recurrence rate. ** *p* < 0.01; * *p* < 0.05. Abbreviations: PD, progressive disease; SD, stable disease; PR, partial response; CR, complete response.

**Figure 6 jpm-12-01840-f006:**
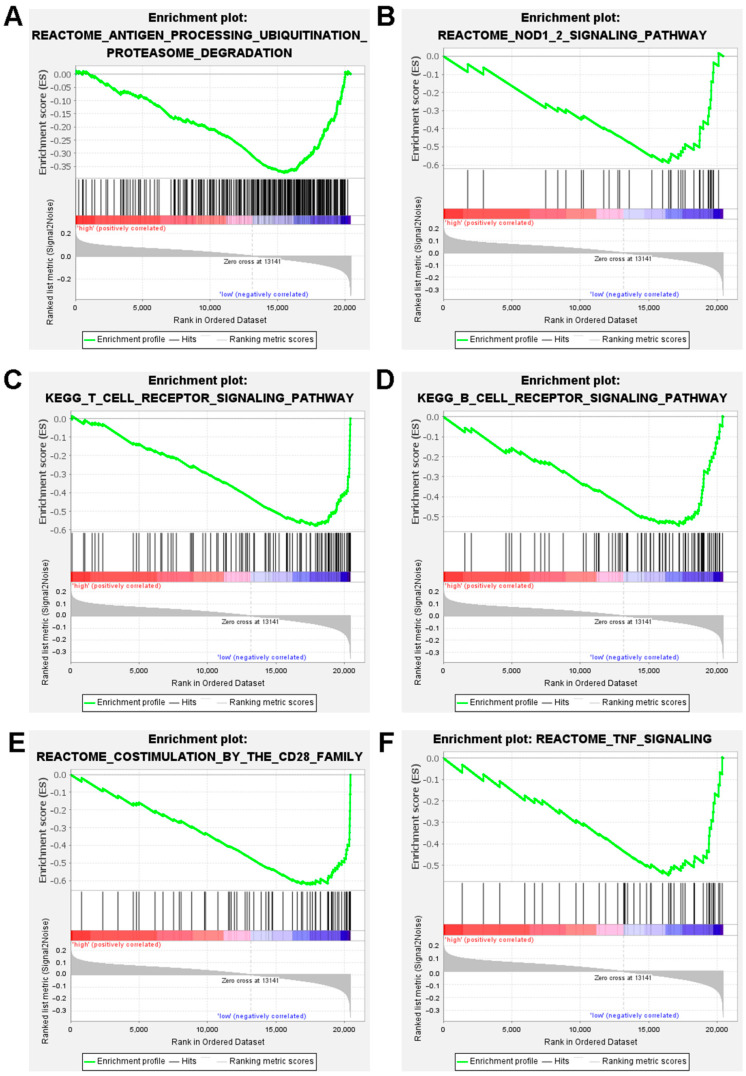
Analysis of the gene set and function enrichment of the ICD-related gene signature in high- and low-risk groups. Antigen processing ubiquitination proteasome degradation pathways (**A**), signaling by NOD 1/2 (**B**), signaling by T cell receptors (**C**), signaling via B cell receptors (**D**), the CD28 family’s costimulation (**E**), and TNF signaling (**F**) are significantly increased in the low-risk group.

**Figure 7 jpm-12-01840-f007:**
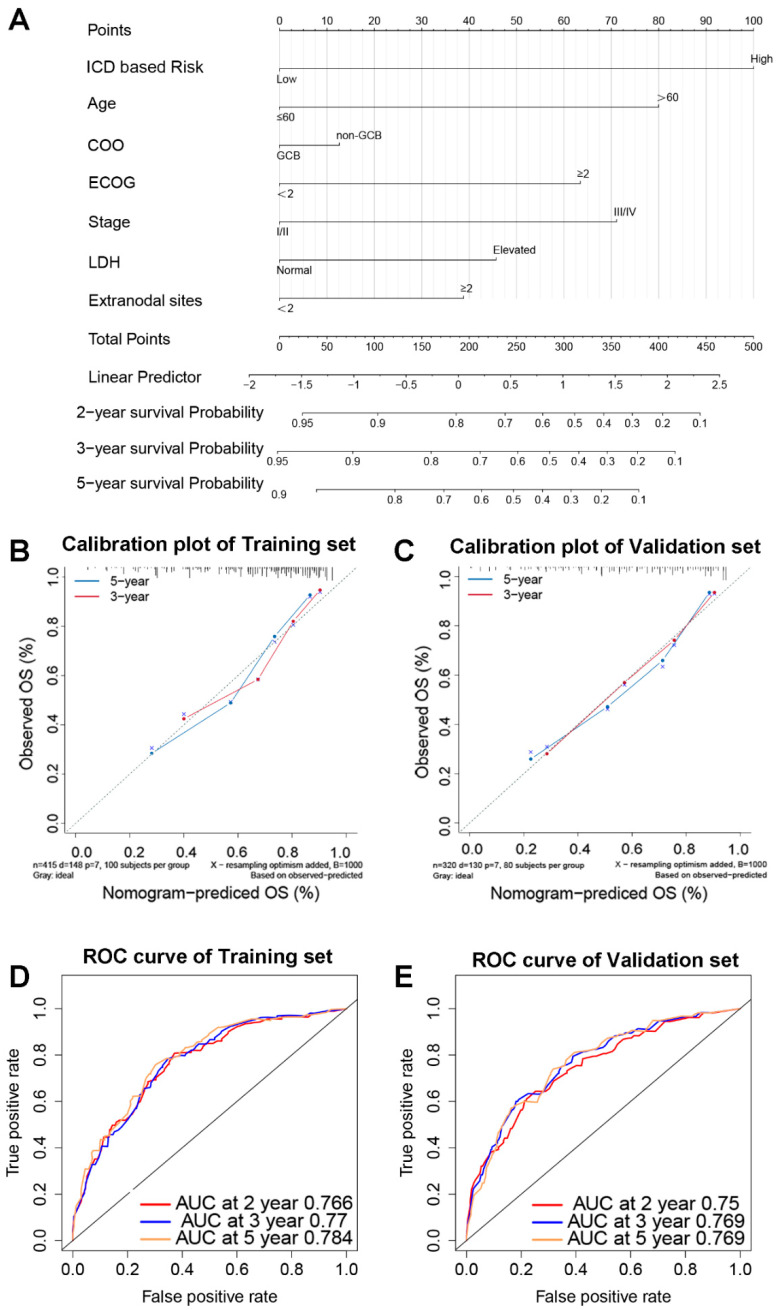
An ICD-related gene signature-based prediction model. The nomogram prediction model for DLBCL patients’ OS (**A**). Calibration graphs of the training set’s 3-year and 5-year survival probabilities (**B**) and the validation set’s (**C**). The 2-year, 3-year, and 5-year OS in the training set (**D**) and validation set (**E**) are predicted by time-dependent ROC curves. Abbreviations: COO, cell of origin; GCB, germinal center B cells; ECOG, Eastern Cooperative Oncology Group; LDH, lactate dehydrogenase; OS, overall survival; AUC, area under the curve.

**Table 1 jpm-12-01840-t001:** Patients’ clinical features from the training set and validation set.

Variables	Training Set GSE31312 (*n* = 415)		Validation Set GSE10846 (*n* = 320)	
	**NO.**	**%**	**NO.**	**%**
Age				
≤60	173	41.7	152	47.5
>60	242	58.3	168	52.5
Gender				
Male	238	57.3	171	53.4
Female	177	42.7	134	41.9
NA	0		15	4.7
ECOG				
1–2	346	83.4	244	76.3
3–4	69	16.6	76	23.8
Ann Anbor stage				
I–II	200	48.2	151	47.2
III–IV	215	51.8	169	52.8
Extranodal sites				
<2	323	77.8	297	92.8
≥2	92	22.2	23	7.2
LDH				
Normal	145	34.9	163	50.9
Elevated	270	65.1	157	49.1
COO				
GCB	206	49.6	142	44.4
Non-GCB	209	50.4	178	55.6

Abbreviations: NA, not available; ECOG, Eastern Cooperative Oncology Group; LDH, lactate dehydrogenase; COO, cell of origin; GCB, germinal center B cells.

**Table 2 jpm-12-01840-t002:** The associations between ICD-related gene signature risk and patients’ clinical features in the training set.

Variables	Low-Risk Group (*n* = 208)		High-Risk Group (*n* = 207)		
	**NO.**	**%**	**NO.**	**%**	** *p* **
Age					0.797
≤60	88	42.3	85	41.1	
>60	120	57.7	122	58.9	
Gender					0.212
Male	113	54.3	125	60.4	
Female	95	45.7	82	39.6	
ECOG					0.141
1–2	179	86.1	167	80.7	
3–4	29	13.9	40	19.3	
Ann Anbor stage					
I–II	108	51.9	92	44.4	0.127
III–IV	100	48.1	115	55.6	
Extranodal sites					0.031
<2	171	82.2	152	73.4	
≥2	37	17.8	55	26.6	
LDH					0.131
Normal	80	38.5	65	31.4	
Elevated	128	61.5	142	68.6	
COO					0.000
GCB	123	59.1	83	40.1	
Non-GCB	85	40.9	124	59.9	

Abbreviations: ECOG, Eastern Cooperative Oncology Group; LDH, lactate dehydrogenase; COO, cell of origin; GCB, germinal center B cells.

## Data Availability

Publicly available datasets were analyzed in this study. The data for this study were downloaded from GEO (http://www.ncbi.nlm.nih.gov/geo/) on 1 May 2021. The data we analyzed are publicly available from the GEO database, and the current research followed the GEO data access policies and publication guidelines.

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
