# Peer review of "Immunogenic Cell Death (ICD)-Related Gene Signature Could Predict the Prognosis of Patients with Diffuse Large B-Cell Lymphoma"

_jpm, 2022, doi:10.3390/jpm12111840_

Round 1

Reviewer 1 Report

I reviewed a manuscript entitled “Immunogenic cell death (ICD)-related gene signature could predict the prognosis of patients with diffuse large B-cell lymphoma.” written by Liqin Ping, et al., which reported immunogenic cell death (ICD)-related gene signature risk status can predict prognosis. In this investigation, comprehensive data set has been analyzed and the results is reasonable. External validation also has been completed and the conclusion is reliable.

Major and minor comments

I did not have any revision request.

Author Response

Dear reviewer:

Thank you for your work and for your kind c­­­­­­omments on our research.

Reviewer 2 Report

I have read the manuscript numbered “JPM-1996756” and entitled “Immunogenic cell death (ICD)-related gene signature could predict the prognosis of patients with diffuse large B-cell lymphoma” with great attention.  The authors here tried to find out the impact of ICD on DLBCL, which I believe very interesting perspective.

The paper was well-written, reader-friendly, and compact. Here are my suggestions for the paper.

·         Introduction is clear and on target.

·     Method section is clear and structured with subheadings and detailed with every stage of the study well mentioned. The validation set supports the findings as well, which is well thought

·         For the patients’ clinical features, were both groups treated similarly? I was not able to see treatment-related data.

·         I believe footnotes for abbreviations should be added for tables and figures.

·         For the univariate and multivariate analysis section, categorical variables included in the model reference are not displayed (such as COO (REF GCB or non-GCB)). I believe for the “Risk” variables the authors are proposing the ICD-related novel risk classification. I believe it may lead to misunderstandings such as IPI etc. It could be stated as “ICD based Risk” etc.

·         For figures 5A and 5B the authors should mention the exact percentage of the response rates.

·         The authors should mention the limitations of the study.

Author Response

Dear reviewer:

Thank you for your kind c­­­­­­omments on our research. We have revised the manuscript carefully according to your valuable comments. The point to point responds to the reviewers’ comments are listed as following:

Comment 1: Introduction is clear and on target.

Response 1: Thank you for your kind comment.

Comment 2: Method section is clear and structured with subheadings and detailed with every stage of the study well mentioned. The validation set supports the findings as well, which is well thought.

Response 2: Thank you for your kind comment.

Comment 3: For the patients’ clinical features, were both groups treated similarly? I was not able to see treatment-related data.

Response 3: Thank you for your helpful comment. The first-line chemotherapy regimen was similar in two datasets; patients were treated with CHOP-like regimen with or without rituximab. The description of the treatment section was added to the manuscript.

Comment 4: I believe footnotes for abbreviations should be added for tables and figures.

Response 4: Thank you for your helpful comment. We have added footnotes for abbreviations for tables and figures in the new manuscript.

Comment 5: For the univariate and multivariate analysis section, categorical variables included in the model reference are not displayed (such as COO (REF GCB or non-GCB)). I believe for the “Risk” variables the authors are proposing the ICD-related novel risk classification. I believe it may lead to misunderstandings such as IPI etc. It could be stated as “ICD based Risk” etc.

Response 5: Thank you for your helpful comment. We have added footnotes for abbreviations for tables and figures in the new manuscript. We describe the reference in the results of cox regression analysis. Meanwhile, we replace "Risk" into "ICD based Risk".

Comment 6: For figures 5A and 5B the authors should mention the exact percentage of the response rates.

Response 6: Thank you for your helpful comment. We have added the exact percentage of the response rates in figures 5A and 5B.

Comment 7: The authors should mention the limitations of the study.

Response 7: Thank you for your helpful comment. We have added the limitations of the study in the manuscript.

Reviewer 3 Report

The authors have analyzed ICD 10 gene signature in diffuse large B cell lymphoma as independent predictor of outcomes. The article is clinically relevant and should be accepted. The only minor revision is to expand a discussion section.

Author Response

Dear reviewer:

Thank you for your kind comments on our research. We have expanded a discussion section as you suggested.